# Identification of Polyphenols in Sea Fennel (*Crithmum maritimum*) and Seaside Arrowgrass (*Triglochin maritima*) Extracts with Antioxidant, ACE-I, DPP-IV and PEP-Inhibitory Capacity

**DOI:** 10.3390/foods12213886

**Published:** 2023-10-24

**Authors:** Marta María Calvo, María Elvira López-Caballero, Oscar Martínez-Alvarez

**Affiliations:** Institute of Food Science, Technology and Nutrition (ICTAN-CSIC), 6th José Antonio Novais St., 28040 Madrid, Spain; mmcalvo@ictan.csic.es (M.M.C.); elvira.lopez@ictan.csic.es (M.E.L.-C.)

**Keywords:** *Crithmum maritimum*, *Triglochin maritima*, halophytes, polyphenols, nutraceutics, antioxidant, dipeptidyl peptidase IV, angiotensin converting enzyme I, prolyl endopeptidase

## Abstract

Sea fennel and seaside arrowgrass are two abundant but underutilized halophytes along the Atlantic and Mediterranean coasts. This study investigated the antioxidant capacity and the potential antihypertensive (Angiotensin Converting Enzyme I, ACE-I inhibition), hypoglycaemic (Dipeptidyl Peptidase IV, DPP-IV inhibition), and nootropic (Prolyl Endopeptidase, PEP inhibition) activity of their polyphenol extracts. They had a high phenol content (21–24 mEq GA/g), antioxidant capacity evaluated using the ABTS (17–2 mg ascorbic acid/g) and FRAP (170–270 mM Mohr’s salt/g) assays, and effective ACE-inhibiting properties (80–90% inhibiting activity at final concentration of 0.5 mg/mL). Additionally, the sea fennel extract displayed high DPP-IV inhibitory capacity (73% at 1 mg/mL), while the seaside arrowgrass extract exhibited potent Prolyl endopeptidase inhibitory capacity (75% at 1 mg/mL). Fractionation by HPLC concentrated the bioactive molecules in two fractions, for which the composition was analyzed by LC-MS/MS. Different chlorogenic acids seemed to play an important role in the bioactivity of sea fennel extract, and different flavonoids, mainly apigenin, luteolin and chrysoeriol, in the bioactivity of the seaside arrowgrass extract. Given their potential health benefits, these extracts could serve as valuable bioactive ingredients and could potentially encourage the cultivation of these species in regions where traditional crops face challenges in growth.

## 1. Introduction

Preserving edible plant species and improving their cultivation is a priority for the agri-food sector. This need is being driven by climate change, which is affecting the entire world. This challenge is being addressed in different ways, such as using varieties that can withstand water stress, increasing the cultivation of drought-resistant species, or those that can be irrigated with high saline water or even seawater. In addition, climate change is increasing land salinity. This problem affects an estimated 1 billion hectares of land, corresponding to 20% of the world’s cultivated soil [1,2], and 50% of the irrigated area [3].

The problems that climate change is causing in crop production can be addressed by introducing or improving the cultivation of halophytes. Halophytes comprise more than 350 species in 20 orders and 256 families, representing 1–2% of the world’s flora [4]. These plants are adapted to environments with salinity levels equal to or higher than 200 mM NaCl and do not require fresh water. This allows them to grow in areas where only saltwater is available. Some species are suitable for human consumption and could be good candidates for large-scale introduction as edible crops, especially in very arid regions where they could be the main (if not the only) feed resource available [1,5,6,7]. Halophytes are not only a good source of nutrients, but also of bioactive compounds. Some species have been used in traditional medicine to treat various diseases [8,9]. Some of the biological activities of halophytes that have been reported include antioxidant, anticholinesterase, anticancer, antimicrobial, anti-inflammatory, analgesic, astringent, diuretic, antipyretic, vasoactive, antibacterial, and cytotoxic [7,10,11].

Sea fennel (*Crithmum maritimum*) is one of the most abundant halophytes in Europe. It is commonly found along the coasts of Southern and Western Europe, and can also be found in other parts of the world [12]. Sea fennel is capable of thriving in salt-free or low saline soils. The nutritional composition of sea fennel was recently described by Sánchez-Faure et al. [13], who reported a high content of calcium and dietary fiber. Its tender leaves and stems are used in salads, pickles, appetizers, and condiments [12,14]. In addition, sea fennel is abundant in bioactive compounds. Recently, extracts from this plant have been used as a functional ingredient to enrich sunflower oil [15]. Sea fennel has also been utilized to obtain essential oils and in traditional medicine [12,16,17,18]. The potential of this plant as a nutraceutical for preventing and treating certain human diseases has been documented due to the presence of antioxidant molecules [19,20], antimicrobials [19,21], antimutagenics [19], vasodilators [20], and cholinesterase inhibitors [20], among others.

Seaside arrowgrass (*Triglochin maritima*) is a common halophyte found in the marshlands of Northern Europe and in the coastal areas of Mediterranean countries. Their nutrient composition has also been recently documented [13]. This plant contains molecules of interest such as fatty acids, amino acids, organic acids, mono- and disaccharides, polyols and polyphenols with health-promoting properties [13,22]. Likewise, there is limited knowledge about the beneficial effects that its consumption could have on human health. The utilization of this plant as a source of bioactive compounds has hardly been reported. A recent study by Boestfleisch and Papenbrock [23] investigated the production of secondary metabolites with antioxidant capacity in seaside arrowgrass under different conditions of soil salinity. They found that the production of these metabolites was increased under conditions of high soil salinity. This suggests that seaside arrowgrass can be used as a source of antioxidants, which are of interest to protect cells from free radical damage.

Halophytes are exposed to high stress situations that result in the production of antioxidant compounds, including polyphenols [24]. Polyphenols do not show significant toxicity or harmful effects on the human body and have promising therapeutic potential [25]. Several polyphenols have the ability to inhibit angiotensin-converting enzyme (ACE), an enzyme linked to the development of hypertension [26,27,28,29,30]. In addition, previous research has described the potential hypoglycemic capacity of polyphenols through their inhibitory effect on dipeptidyl peptidase IV (DPP-IV) [28,31,32,33]. Additionally, some polyphenols are effective inhibitors of prolyl endopeptidases (PEP) [28,34,35,36], a family of enzymes that can hydrolyze hormones such as vasopressin, thyrotropin-releasing hormone, or substance P. Some researchers have suggested that PEPs may present a viable pharmacological target for treating various neurological conditions linked to abnormally high serum PEP activity, including Alzheimer’s disease, schizophrenia, and depression, as well as other disorders like anorexia and bulimia [37].

In certain areas where salinization of agricultural land and freshwater shortages are reducing crop yields, halophytes offer an alternative to traditional crops. Many studies have focused on evaluating the growth and yield of halophyte crops, as well as the impact of their cultivation on soil and water quality. Others have focused on characterizing their nutritional profile and the study of their potential for animal and human consumption. However, their potential as a potent and natural source of bioactive molecules is poorly documented. In this context, this study aims to explore the potential of two halophytes abundant in the European and Mediterranean coasts, seaside arrowgrass and sea fennel, as a source of molecules of nutraceutical interest against diseases of high prevalence in developed countries. More specifically, this study is focused on the analysis of the antihypertensive, nootropic, hypoglycemic and antioxidant potential of ethanolic extracts of these two plants, as well as on the isolation and identification of the polyphenols potentially responsible for these activities.

## 2. Materials and Methods

### 2.1. Chemicals

Folin–Ciocalteu reagent, gallic acid (GA), 2,2′-azino-bis-(3-ethylbenzothiazoline-6- sulphonic acid) (ABTS), 2,4,6-Tris(2-pyridyl)-s-triazine (TPTZ), ammonium iron (II) sulphate, human dipeptidyl peptidase-IV, angiotensin-converting enzyme from rabbit lung, and reagents used for HPLC analysis were purchased from Sigma-Aldrich, Co. (St. Louis, MO, USA). Seikagaku Corp. (Tokyo, Japan) provided prolyl endopeptidase from *Flavobacterium.* Chromogenic substrates were purchased from Bachem (Bubendorf, Switzerland). Ethylenediaminetetraacetic acid (EDTA) was from Leco Corp. (St. Joseph, MI, USA). Analytical-grade chemicals and reagents were purchased from Panreac Chemical Co. (Barcelona, Spain).

### 2.2. Plant Processing

The company Porto-Muiños S.L. (Cedeira, A Coruña, Spain) collected sea fennel and seaside arrowgrass on the coast of Galicia, in north-western Spain. The samples were transported at a low temperature (4 °C) to Madrid. The edible portions (leaves and stems) underwent a washing process involving distilled water and then were subjected to drying via forced air oven (FD 240, Binder, Tuttlingen, Germany) at 55 °C for 24 h. The dried plants were then stored at 4 °C until analysis.

### 2.3. Preparation of the Extract

The preparation of the extract was in accordance with the method of Calvo et al. [28]. The dried samples (250 g) were homogenized in a 400 mL mixture of ethanol/water 1/1 (*v*/*v*) at pH 2, in a ratio of 1:2 (*w*/*v*) using an Ultra-Turrax homogenizer (Mod. T25D, IKA^®^-Werke GmbH & Co. KG, Staufen, Germany) at 25,000× *g* for 3 min at 25 °C. The homogenates were kept in an ice bath and then sonicated in 16 cycles at 90% amplitude, with 60 s of interval between each cycle of 1 min. After extraction, the samples were centrifuged at 12,000× *g* for 10 min at 5 °C (Sorvall evolution, Thermo Fisher Scientific, Waltham, MA, USA). The supernatants were evaporated using a rotary evaporator (R-300, BÜCHI, Buchegg, Switzerland) until the ethanol was eliminated. Then, the extracts were freeze-dried and stored at 4 °C until analysis.

### 2.4. Determination of Total Phenol Content

The total phenolic content of the extracts was determined using the Folin–Ciocalteau assay, in accordance with the procedure outlined by Calvo et al. [28]. The extracts were dissolved in distilled water to a concentration of 10 mg/mL (dry weight). The extracts were then mixed (1/1, *v*/*v*) with 500 µL of Folin reagent, 10 mL of 0.7 M Na_2_CO_3_, and 14 mL of distilled water. The mixtures were stirred for 1 h at room temperature, and the absorbance was then measured at 750 nm using a spectrophotometer Shimadzu UV-1601 (Kyoto, Japan). Gallic acid (0.05–0.3 mg/mL) was used as standard. The results were expressed as mg Eq GA/g (dry weight).

### 2.5. Scavenging Activity of ABTS Radical Cation

The radical scavenging activity of the fractions and the extract was performed according to Calvo et al. [28]. First, an ABTS+ stock solution was prepared by mixing 7 mM aqueous ABTS+ and 2.45 mM K_2_O_8_S_2_ in a 1:1 (*v*/*v*) ratio. After 16 h in the dark, the mixture was diluted in distilled water to reach an absorbance of 0.70 ± 0.02 at 734 nm. Then, 980 µL of ABTS+ working solution was mixed with 20 µL of sample diluted in distilled water (10 mg of extracts/mL or 1 mg of fraction/mL, dry weight). The mixture was incubated for 10 min at 30 °C in the dark. The absorbance was then measured at 734 nm using a spectrophotometer. Ascorbic acid (0.02–0.6 mg/mL) was used as standard. The results were expressed as mg Eq ascorbic acid/g (dry weight).

### 2.6. Ferric Reducing Antioxidant Power (FRAP)

The ferric-reducing antioxidant power of the fractions and the extract was analyzed following the protocol of Calvo et al. [28]. To prepare the FRAP reagent, 25 mL of 0.2 M sodium acetate buffer (pH 3.6), 2.5 mL of 10 mM TPTZ (dissolved in 40 mM HCl), and 2.5 mL of 20 mM FeCl_3_ were mixed. Afterwards, 900 µL of FRAP reagent was combined with 90 µL of distilled water and 30 µL of sample (10 mg of extract or 1 mg of fraction/mL, dry weight), and incubated at 37 °C for 30 min in the dark. The absorbance was measured at 595 nm using a spectrophotometer. Ammonium iron (II) sulphate, also known as Mohr’s salt, was used as a standard at concentrations ranging between 0.1 to 1 mM. The results were expressed as mEq Mohr’s salt per gram of dry weight.

### 2.7. Determination of ACE Inhibitory Activity

The ACE inhibitory activity was measured according to Sentandreu and Toldrá [38] with some modifications. To prepare the enzyme, it was dissolved in a Tris-base buffer (150 mM, pH 8.3, containing 1.125 M NaCl). The final enzymatic activity was 15 mU/mL. The samples (final concentration of 0.5 mg of dried extract/mL or 0.1 mg of dried fraction/mL in the system) and the substrate (0.45 mM Abz-Gly-Phe(NO_2_)-Pro) were also dissolved in the same buffer. A mixture consisting of 50 µL of either sample or buffer (control) and 50 µL of ACE was incubated at 37 °C for 15 min. The reaction was started by adding 200 µL of substrate and continued for 15 min. The fluorescence was measured at λexc 360 nm and λem 400 nm using a Sinergy Mx microplate reader (BioTeck, Colmar, France). The maximum linear increase in fluorescence per minute was determined for each well and inhibition activity was expressed as a percentage.

### 2.8. Determination of PEP Inhibitory Activity

PEP-inhibitory activity was measured in 96-well plates, as according to Sila et al. [35]. In each well, 20 μL of enzyme (1 mU) was mixed with either 180 μL of 0.1 M sodium phosphate buffer pH 7 (assay buffer) or 150 μL of buffer, together with 30 μL of the sample diluted in buffer. The final concentration of sample in the system was 1 mg of dried extract/mL or 0.2 mg of dried fraction/mL. The plate was incubated at 37 °C for 15 min. Then, 100 μL of 0.01 mM Z-Gly-Pro-AMC was added. The microplate reader monitored fluorescence at λexc 340 nm and λem 450 nm every 1 min for 20 min. The inhibitory activity was expressed as a percentage of the inhibition, as outlined in Section 2.7.

### 2.9. Determination of DPP-IV Inhibitory Activity

The DPP-IV-inhibition was determined as described by PEP-activity, but with 0.1 M Tris-HCl (pH 8) as buffer assay, following the protocol outlined by Sila et al. [35]. The substrate used was 0.025 mM H-Gly-Pro-AMC·HBr. The inhibiting activity of the samples was calculated as indicated in Section 2.7.

### 2.10. Chromatographic Fractionation of the Extracts

The extracts were fractionated following the method of Calvo et al. [28]. The extracts were firstly reconstituted to a concentration of 80 mg/mL (dry weight) in a mixture of ethanol and water (1/1, *v*/*v*). A preparative HPLC system (Agilent LC PREP 1260 Infinity Series, Santa Clara, CA, USA) equipped with a diode array detector (DAD) and a Tracer Excel 120 ODS-A 25 × 0.78 cm preparative C18 column (Teknokroma, Barcelona, Spain) was used. Phase A was comprised of ultrapure water containing 5% acetonitrile and 0.1% formic acid, while phase B consisted of acetonitrile with 0.1% formic acid. The gradient used consisted of four steps: 0–15% B for the first 15 min, 15–50% B for the following 5 min, 50–65% B for the next 15 min, and finally 65–0% B for the last 10 min. The flow rate was maintained at 2 mL/min. Fractions were automatically collected using a fraction collector while detection was performed at 280 and 360 nm. The collected fractions underwent solvent evaporation in a Speed Vac concentrator (Thermo Fisher, Waltham, MA, USA). Then, the fractions were lyophilized and stored at 4 °C until use.

### 2.11. Tentative Identification of Polyphenols by HPLC-ESI-QTOF-MS

The fractions showing the highest bioactivity were selected and the polyphenols were extracted using Oasis^®^ HLB cartridges (Waters, Milford, MA, USA). The separation of the polyphenols was performed using high-performance liquid chromatography coupled with electrospray ionization and quadrupole time-of-flight mass spectrometry (HPLC-ESI-QTOF-MS) as described by Sánchez-Faure et al. [13]. The mass spectra were obtained by electrospray ionization in negative mode. The gas temperature was 325 °C and the drying gas flow was 12 L/min. A range of 100 to 1200 m/z was used to scan the acquisition for auto MS/MS. The mass spectra were analyzed with MassHunter Workstation version 4.0 software (Agilent Technologies, Santa Clara, CA, USA). The phenolic compounds were identified by comparing their experimental mass with the mass obtained in the negative mode analysis, allowing an error of 8 ppm. The fragmentation pattern and the relative abundance of fragment ions obtained for each compound were also compared with some databases such as MassBank, Food Database and Human Metabolome Database.

### 2.12. Statistical Analysis

A student’s *t*-test was used to compare the bioactivity of the extracts from sea fennel and seaside arrowgrass. The bioactivity of the fractions was compared using the one-way analysis of variance (ANOVA) method. The software program SPSS 27 (IBM Corporation, Armonk, NY, USA) was used to perform the statistical analysis. The level of significance was set at *p* ≤ 0.05. All measurements were taken three times to ensure accuracy.

## 3. Results and Discussion

### 3.1. Phenolic Content and Bioactivity of the Extracts

#### 3.1.1. Phenolic Content

The phenolic content of the sea fennel and seaside arrowgrass extracts was 23.6 and 21.2 mg Eq GA/g, respectively (Table 1). According to Kähkönen et al. [39], these plants can be considered rich in phenols, as they have values above 20 mg Eq GA/g. The high phenol content in halophytes has been previously described by several authors, such as Lopes et al. [9]. Their synthesis and accumulation are a response to abiotic stress situations such as extreme temperatures, salinity, sodicity and alkalinity of soils, or flood and anoxia situations. The phenol content can also be increased by exposure to biotic threats [9].

The phenolic content found in the sea fennel extract was higher than that reported by Houta et al. [40] (11.5 mg GA/g) and Sánchez-Faure et al. [13] (3.6 mEq GA/g). Kraouia et al. [12] reviewed the phenolic content reported by different authors for sea fennel extracts obtained with different solvent mixtures (acetone, methanol or ethanol with water) and, in almost all cases, the values were lower than that obtained in this work. Extracts with a slightly higher phenol content have been obtained using methanol/water (up to 33 mg GA/g) [41] and ethanol/water and microwaves (26–29 mg GA/g) [42]. Concerning the phenolic content of seaside arrowgrass, it was higher than those reported by Lee et al. [22], 3.3 mg GA/100 g, and Sánchez-Faure et al. [13], 5.9 mg GA/g. All these differences can be attributed to different factors, such as the stress conditions, the habitat, the organ selected for extraction, the genotype, the vegetative state, the extraction parameters and the polarity indexes of the different solvent mixtures used [9,12,13].

#### 3.1.2. Antioxidant Activity

ABTS and the FRAP assays were used to determine the antioxidant activity. Both methods are based on electron transfer. The ABTS assay measures the ability of a compound to scavenge free hydrophilic radicals, while the FRAP assay measures the ability of a compound to donate electrons to free radicals, converting them into more stable products and thus terminating the free radical chain reaction [43].

The antioxidant activity of the sea fennel extract was slightly higher than that of the seaside arrowgrass extract in both assays. The ABTS assay results ranged from 17.4–20.6 mg Eq ascorbic acid/g, while the FRAP assay results ranged from 170–269 mEq Mohr salt/g. Kadoglidou et al. [44] and Meot-Duros and Magné [41] also obtained sea fennel extracts with radical scavenging activity. Nabet et al. [45] reported a lower antioxidant capacity, with values of 0.43 mg TE/g. Lee et al. [22] determined the antioxidant capacity of ethanolic extracts of six halophytes using the DPPH assay, and found that seaside arrowgrass had similar or higher antioxidant activity than different Salicornia species. The content of antioxidant enzymes, phenolic compounds, carotenoids, and chlorophylls could be responsible for the potential antioxidant activity of the sea fennel and the seaside arrowgrass extracts. However, phenolic compounds appear to be a primary factor in the antioxidant activity of the extracts, according to Zengin et al. [46], who noted their effects on radical scavenging and reducing power mechanisms in seven Apiaceae species.

#### 3.1.3. Enzyme Inhibitory Capacity of the Ethanolic Extracts

The ACE inhibitory capacity of sea fennel and seaside arrowgrass extracts was remarkable, especially for the seaside arrowgrass extract, with values of around 90% inhibition at a final concentration of 0.5 mg/mL (Table 1). The ability of halophyte extracts to inhibit enzymes involved in various human pathologies, such as ACE, has been poorly described. Calvo et al. [28] described the ACE inhibitory capacity of an ethanolic extract of the halophyte ice plant, reaching a high inhibition (90.5%) but at higher final concentration (1 mg/mL) than that tested in this work. Loizzo et al. [47] reported an important ACE inhibitory activity of different ethyl acetate extracts of several halophytes belonging to the genus Salsola (IC_50_ values among 0.18–0.28 mg/mL).

The ACE inhibitory activity of halophyte plant extracts could be associated with the presence of certain terpenoids, alkaloids, tannins, proanthocyanidins and flavonoids. Men et al. [48] attributed the ACE inhibitory capacity of different extracts obtained from *Suaeda physophora* to the presence of polyphenols, although this capacity was much lower than that of the sea fennel and seaside arrowgrass extracts. Shukor et al. [49] investigated the ACE inhibitory capacity of phenolic compounds, observing different modes of action. The antihypertensive capacity of polyphenols has also been studied *in vivo* [50], with this effect attributed not only to their ability to inhibit ACE, but also to other mechanisms [50,51]. However, information regarding the *in vivo* antihypertensive effect of halophyte extracts is limited. Sharifi et al. [52] and Phillips et al. [53] observed an antihypertensive effect of Gokshur (*Tribulus terrestris*) extracts in experimental animals, which was attributed to their ability to inhibit ACE as well as the release of nitric oxide.

The extracts showed high DPP-IV inhibiting capacity (68–73%) with a significantly higher potency of the sea fennel extract (Table 1). Some authors have observed the hypoglycemic activity of halophyte extracts in vivo, but this has been attributed to mechanisms other than DPP-IV inhibition [54].

The seaside arrowgrass and sea fennel extracts showed a high PEP-inhibiting capacity, especially the former (Table 1). This capacity, although high, was lower than that described for an ethanolic extract of the halophyte ice plant at the same final concentration of 1 mg/mL (98%) [28]. The inhibition of PEP by halophyte extracts has received limited description, and may be associated with specific polyphenols, primarily flavonoids, as well as cyclotides and alkaloids [55,56,57].

### 3.2. Characterization and Bioactivity of the Fractions

The sea fennel extracts therefore displayed significant potential as antihypertensive, hypoglycemic and antioxidant agents (Table 1). To isolate and identify the molecules potentially responsible for the bioactivity, the extracts were chromatographically fractionated, and the potentially bioactive activity of each fraction was further analyzed.

Each extract was separated into four fractions (Figure 1 and Figure 2). Table 2 revealed a strong correlation between the bioactive potential and phenol content in both extracts, suggesting that the bioactive molecules in each extract were likely polyphenols. The molecules with the strongest ACE inhibitors and antioxidants were concentrated in SF-3, SF-4, SA-3, and SA-4. SF-3 and SF-4 concentrated the most significant DPP-IV inhibitors, and SA-3 and SA-4 the most potent PEP inhibitors. These four fractions were chosen to determine their polyphenol composition and role in the analyzed bioactive activities.

#### 3.2.1. Identification of Potential Bioactive Polyphenols Present in the Factions

The polyphenolic composition of SF-3 and SF-4 was similar, as shown in Table 3. Some of the identified polyphenols were also found in an extract of sea fennel collected from the same coast [13], using sequentially ethanol/water (1/1, *v*/*v*) and acetone/water (7/3, *v*/*v*) as solvent extraction mixtures. Both SF-3 and SF-4 exhibited a high concentration of hydroxycinnamic acids, mainly isochlorogenic acids (Table 4). This was especially remarkable in SF-3, where 91% of the total polyphenols were hydroxycinnamic acids. This fraction predominantly consisted of caffeoylquinic acids (41.7%), coumaroylquinic acids (30.2%), and feruloylquinic acids (12.6%), which accounted for approximately 84–85% of the total polyphenols identified. SF-4 contained less hydroxycinnamic acids (67.9%), with coumaroylquinic acids (29.9%), feruloylquinic acids (13.26%) and dicaffeoylquinic acids (13.12%) contributing to 56.4% of the total polyphenols in the fraction. Other authors have also identified chlorogenic acids as the primary components in sea fennel extracts [12,16,19,20,41,42,58,59]. The caffeoylquinic acids present in SF-3 and SF-4 might correspond to 1, 3, 4 and 5-caffeoylquinic acids. Kraouia et al. [12] also reported the presence of major isomers of caffeoylquinic acid in extracts from leaves and stem tissues of sea fennel collected from various regions. Additionally, SF-3 and SF-4 contained four isomers of dicaffeoylquinic acid (mainly in the latter, 13.12%), but at a low relative concentration. Four coumaroylquinic acid isomers (2, 3, 4 and 5-coumaroylquinic acids, around 30%) and two feruloylquinic acid isomers (around 13%) were also found in both fractions. Furthermore, SF-4 contained approximately 31% of flavonoids, predominantly quercetin glycosides (12.15%) and rutin (7.18%). Rutin has been previously found in sea fennel extracts [16,42], as well as quercetin-3-O-galactoside [16]. Furthermore, SF-3 and SF-4 contained other polyphenols (Table 3 and Table 4), but their relative content was minimal.

The polyphenolic profiles of seaside arrowgrass fractions SA-3 and SA-4 differed significantly from those of SF-3 and SF-4 (Table 5). Both SA-3 and SA-4 consisted almost entirely of flavones and derivatives, but there were notable differences in their composition. SA-3 was mainly composed of apigenin 6-hexoside (80.5% of the total), followed by luteolin (8.5%) (Table 6). In contrast, SA-4 was mainly composed of apigenin (52.4%), luteolin (34.7%) and chrysoeriol (12.8%). Sánchez-Faure et al. [13] analyzed the polyphenol profile of a seaside arrowgrass extract harvested in the same area, and discovered a predominance of flavones and flavonols. However, they observed a significant abundance of isoorientin (a glycosylated luteolin) in the extract.

#### 3.2.2. Antioxidant Activity of the Fractions

Fractions SF-3, SF-4, SA-4 and SA-3 (in that order) showed the most potent antioxidant activity (Table 2), as determined by the ABTS or FRAP assays. Table 2 demonstrates a positive correlation between the antioxidant capacity of the fractions and their phenol content, as observed by Lee et al. [22], suggesting that phenolic compounds were the main compounds responsible for the antioxidant activity.

The antioxidant activity of SF-3 and SF-4 was mainly ascribed to the presence of chlorogenic acids. It was also observed by Siracusa et al. [60] in other sea fennel extract. The hydroxyl groups on the aromatic ring (Figure 3) have been associated with the antioxidant properties of these hydroxycinnamic acids, reflecting the facilitating of hydrogen atom donation, which effectively inhibits radical chain reactions and other oxidants, and the ability to act as chelators of transition metals, preventing Fenton-type processes [61,62]. Xu et al. [61] compared the antioxidant activity of different chlorogenic acid isomers. They found that the dicaffeoylquinic acids had better antioxidant activity than caffeoylquinic acids due to additional OH groups being attached to the aromatic ring in their structure. Moreover, flavonoids such as flavonols, flavones, and their derivatives, primarily rutin, may play a significant role in the antioxidant activity of SF-4 [63], as also observed by Siracusa et al. [60]. Myricetin, though present in a low relative concentration, could also contribute to the antioxidant activity of SF-4 [64].

SA-4 and SA-3 exhibited considerable antioxidant activity, but it was lower than that of SF-3 and SF-4 (Table 2). The antioxidant activity of SA-3 and SA-4 was attributed to the abundance of flavones and derivatives.

SA-4 exhibited a significant greater level of antioxidant activity compared to SA-3, which may be due to its higher polyphenol content and/or its relatively higher levels of apigenin, luteolin and chrysoeriol content. The antioxidant activity of flavonoids depends on the number and position of OH groups within their structure, as well as the presence of electron-withdrawing or electron-donating substituent groups [62,65]. The antioxidant activity of apigenin has been recently reviewed by Kashyap et al. [66], and has been demonstrated *in vitro* and in animal trials. Apigenin is a potent inhibitor of several oxidant enzymes, including cyclooxygenase, nitric oxide synthase, xanthine oxidase, nitric oxide, and lipoxygenase. Additionally, it can interact with redox signaling pathways, and scavenge free radicals through two pathways: hydrogen atom transfer (HAT) and single electron transfer (SET). This effect is due to OH groups present in the structure at the 4′ position of the phenolic B-ring and the 5 and 7 positions of the A-ring [67] (Figure 4). These groups donate their hydrogen atoms and electrons to the hydroxyl, peroxyl and peroxynitrite radicals, rendering them stable and generating stable radicals. The presence of a C2-C3 double bond and a carbonyl group at the 4 position of the pyranyl C-ring (1,4 pyrone moiety) enhances the antioxidant activity by enabling the delocalization of unpaired electrons across the A-, B- and C-rings. It leads to the formation of a more stable phenoxyl radical. Similarly, the antioxidant activity of chrysoeriol (an apigenin with a methoxy group in 3′ position of the B-ring) can also be attributed to these same groups. Regarding luteolin, its antioxidant capacity is due to the presence of the 1,4-pyrone moiety and OH groups, including an additional one in the 3’ position of the B-ring (Figure 4) [68].

#### 3.2.3. Enzyme Inhibitory Activities of the Bioactive Fractions

##### ACE-Inhibitory Activity

SF-3, SF-4, SA-3 and SA-4 displayed potent ACE inhibitory capacity (Table 2). Significant differences were observed among samples, showing SF-E the most potent inhibiting effect. The considerable content of chlorogenic acids in SF-3 and SF-4 (especially in SF-3) indicates their vital contribution to the ACE-inhibitory potential of the sea fennel extract. Agunloye et al. [69] observed that intraperitoneal administration of caffeoylquinic acid in hypertensive rats resulted in a lowering of blood pressure and ACE-inhibiting activity. The reduction in ACE activity was attributed to a decrease in vascular ACE synthesis and secretion, or to ACE inhibition. The ACE-inhibitory activity of the chlorogenic acids could be due to the OH groups in the benzene ring, which interact with zinc ions and disulfide bridges at the active site of ACE [70,71]. Thus, dicaffeoylquinic acids, with four OH groups on the benzene rings, display significant ACE inhibition, in contrast to coumaroylquinic acids, with only one OH group. Moreover, the carboxylate moieties in the hydroxycinnamic acids enhance ACE inhibition by promoting a charge–charge interaction with the zinc ion at the ACE active site via the oxygen atom [49]. In contrast, methylation of an OH group, as observed in 3-feruloylquinic acid, decreases the ACE inhibitory capacity through steric hindrance that obstructs the binding to the ACE active site [72]. Flavones, flavonols and their derivatives (especially quercetin, rutin and apigenin) in SF-3 and SF-4 also exhibit a certain ACE-inhibitory activity [49]. However, despite their greater presence in SF-4, it did not result in a higher bioactivity when compared with SF-3. Quercetin hexosides, primarily present in SF-4, may also participate in ACE-inhibition in a competitive manner [25]. The available evidence suggests that the polyphenols in the fractions synergistically inhibited ACE, but certain polyphenols, particularly the dicaffeoylquinic acids, had a greater inhibitory effect than others.

The ACE inhibitory effect of SA-3 and SA-4 may be due to the significant presence of flavones and their derivatives, with SA-4 demonstrating higher inhibiting activity than SA-3. This difference in activity may be attributed to the relatively higher abundance of apigenin, luteolin, and chrysoeriol (Table 6). Luteolin has been identified as a flavonoid with antihypertensive properties in rat models, using mechanisms other than ACE inhibition [50], although it cannot be ruled out that it could exert this effect *in vivo*. Loizzo et al. [47] reported the ACE-inhibiting activity of various flavonoids, such as apigenin and luteolin, with similar IC_50_ values obtained for both (280 and 290 µM, respectively). Jenis et al. [73] also reported the ACE-inhibiting activity of apigenin and luteolin, extracted from *Limonium michelsonii*. Shukor et al. [49] compared the ACE-inhibiting activity of different flavonoids and found that the inhibitory effect of apigenin was lower than that of other flavonoids such as quercetin or kaempferol. As previously indicated, the presence of OH groups in the benzene ring is important to inhibit ACE by forming chelate complexes with the zinc ion or certain amino acids within the active center. In addition, the presence of a catechol group in the B-ring of polyphenols, as in the case of luteolin, has been described as an important factor in their ACE inhibitory capacity [74]. Parellada et al. [70] reported that a planar structure essential for the inhibition of metalloenzymes is conferred by the presence of a 2,3 double bond that conjugates the benzene A and B rings. To exert ACE inhibition, the three most abundant flavones identified satisfy the condition of having at least one OH group on each aromatic ring and a carbonyl group in position 4. Chrysoeriol has also been reported as ACE-inhibitor [75]. However, Geng et al. [72] indicated that its inhibitory activity may be reduced due to the methylation in the OH group at the 3′-position of the B-ring.

##### DPP-IV Inhibitory Activity

The high DPP-IV inhibitory capacity of SF-3 and SF-4 (Table 2) can be attributed to the presence of hydroxycinnamic acids. Although the two fractions had similar polyphenolic profiles, their concentration was different. SF-3 exhibited a significantly higher DPP-IV inhibitory activity and a significantly higher proportion of caffeoylquinic acid isomers. These compounds are potent DPP-IV inhibitors. In a recent study, Geng et al. [72] described the DPP-IV inhibitory capacity of 3- and 4-caffeoylquinic acids, and suggested that these compounds may have hypoglycaemic effects *in vivo*. The inhibitory effect of these compounds, as well as other hydroxycinnamic acids found in SF-3 and SF-4, occurs via competitive mechanisms. They bind to the active site by forming hydrogen bonds between their OH groups and (i) the OH groups of the serine side chain present in the active center, (ii) the NH_2_ group of an Arg of the active center and (iii) the C=O groups of Glu side chains [31,76]. Due to steric hindrance caused by the additional caffeoyl group, the dicaffeoylquinic acids found in the fractions, mainly in SF-4, have a weaker inhibitory effect. This obstacle prevents the formation of hydrogen bonds and hinders interaction with the active site of the enzyme [77]. Caffeic acid, found in both fractions at low levels, has been identified as a feeble inhibitor of DPP-IV. Flavonoids, present in both fractions but in low proportions, may have a role to play in DPP-IV inhibition. Thus, apigenin and luteolin have been reported as potent non-competitive DPP-IV inhibitors. These compounds inhibit DPP-IV through the formation of hydrogen bonds and pi interactions with specific residues within the DPP-IV active center [31]. In contrast, quercetin, present in SF-4 but at a very low concentration, exhibited weak inhibition of DPP-IV. Rutin lacks the ability to inhibit DPP-IV due to two sugar groups present in its structure that could impede steric binding to the active site [31]. All of these findings suggest that caffeolquinic acids are the main responsible for the DPP-IV inhibiting capacity of SF-3 and SF-4.

##### PEP Inhibitory Activity

SA-3 and SA-4 showed potent PEP-inhibitory activity at very low concentrations in the system (0.2 mg/mL). The ability of polyphenols to induce PEP inhibition has been scarcely described. Calvo et al. [28] reported the PEP inhibitory activity of an ice plant extract containing luteolin. This compound was detected in both SA-3 and SA-4, but in much greater abundance in SA-4 (Table 6), suggesting that this compound is likely the main one responsible for the PEP inhibitory activity [78]. The presence of two OH groups on the B-ring (catechol ring) of luteolin (Figure 4), and an OH group at the 7-position of the A-ring appears to be linked to the PEP-inhibiting mechanism [31,78]. Apigenin might also play a role in the inhibitory effect of the fractions, either as a free form or in the methoxylated and glycosylated forms. However, according to Lee et al. [78], luteolin has a higher PEP inhibitory activity compared to apigenin (whether in free form or glycosylated), suggesting that luteolin is responsible for the PEP inhibitory capacity of the fractions obtained from seaside arrowgrass extract.

## 4. Conclusions

Seaside arrowgrass and sea fennel are sources of polyphenol extracts that possess antioxidant capacity and have the potential to provide antihypertensive, hypoglycemic and nootropic benefits. The antioxidant and the potential antihypertensive and hypoglycemic properties of sea fennel extract are due to the presence of hydroxycinnamic acids, particularly chlorogenic acids. The antioxidant and the potential antihypertensive and nootropic properties of seaside arrowgrass are attributed to the presence of flavones and derivatives, mainly apigenin and luteolin. These plants are abundant along the Mediterranean and Atlantic coasts and thrive in saline soils, and their potential as a source of bioactive extracts could stimulate greater interest in their controlled cultivation and the development of agriculture in countries with limited arable land. The polyphenols present in these plants can be readily extracted and utilized as functional ingredients in a variety of foods. Their antioxidant properties also make them valuable assets in safeguarding lipids in food products from oxidation. Further work will be required to enhance polyphenol extraction and evaluate the impact of incorporating these compounds into food on their sensory properties. Also, it will be necessary to evaluate the effects of thermal processing on the bioactivity, to assess the bioavailability of the bioactive polyphenols after gastrointestinal digestion, as well as to analyze the quantity and composition of the extracts obtained from plants gathered at different seasons and locations. Finally, it will be essential to evaluate the cytotoxicity of these extracts to determine their potential as bioactive ingredients in functional foods.

## Figures and Tables

**Figure 1 foods-12-03886-f001:**
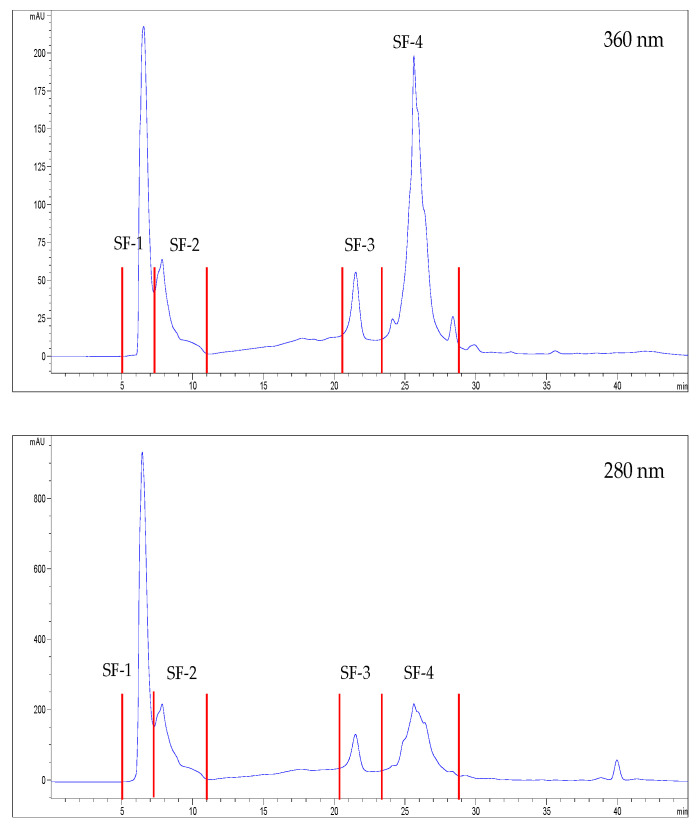
Elution profile of the sea fennel extract measured at 360 and 280 nm. Four fractions were collected (red lines): SF-1 (5–7.2 min), SF-2 (7.3–11 min), SF-3 (20.5–23.4 min), SF-4 (23.7–29 min).

**Figure 2 foods-12-03886-f002:**
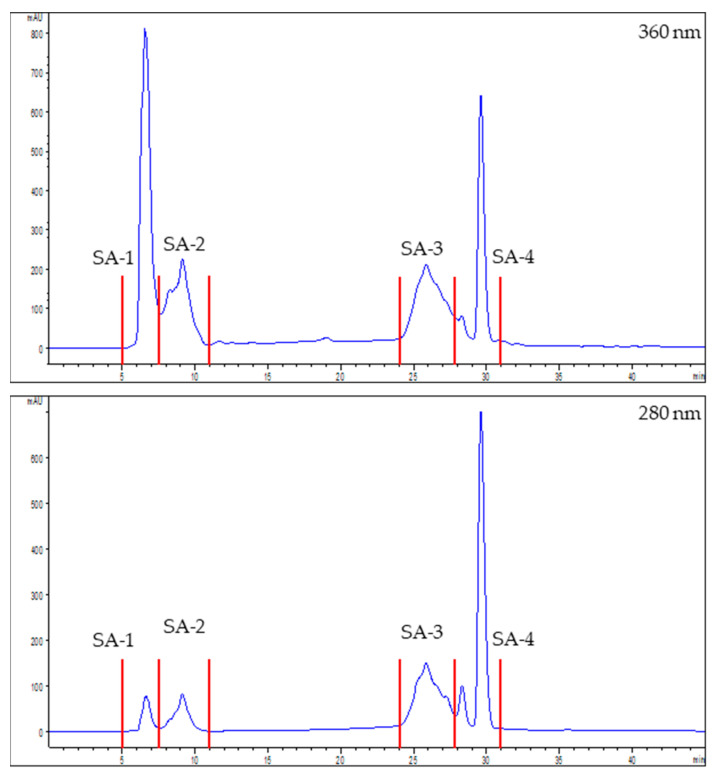
Elution profile of the seaside arrowgrass extract measured at 360 and 280 nm. Four fractions were collected (red lines): SA-1 (5–7.3 min), SA-2 (7.5–11 min), SA-3 (24–27.5 min) and SA-4 (28–31 min).

**Figure 3 foods-12-03886-f003:**
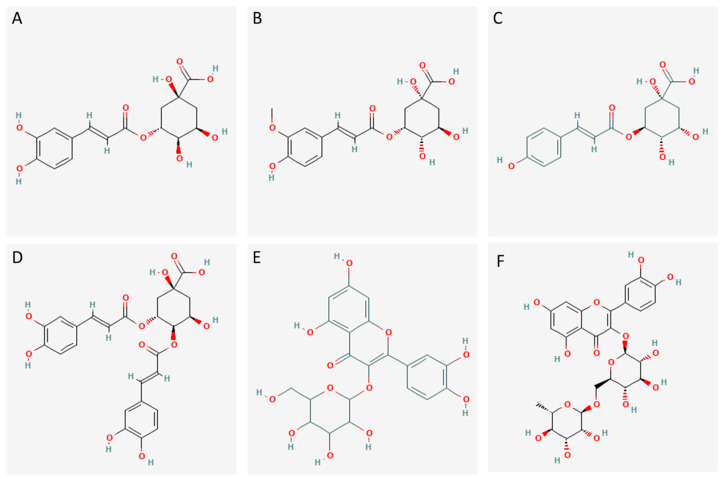
Structure of the most abundant polyphenols in the bioactive fractions from the sea fennel extract. (**A**) 3-Caffeoylquinic acid; (**B**) 3-Feruloylquinic acid; (**C**) 3-Coumaroylquinic acid; (**D**) 3,4-dicaffeoylquinic acid; (**E**) Quercetin 3-O-hexoside (isoquercitrin); (**F**) Rutin. The structures in (**A**–**E**) correspond to one of the isomers potentially present in the fractions. Structures were from PubChem database.

**Figure 4 foods-12-03886-f004:**
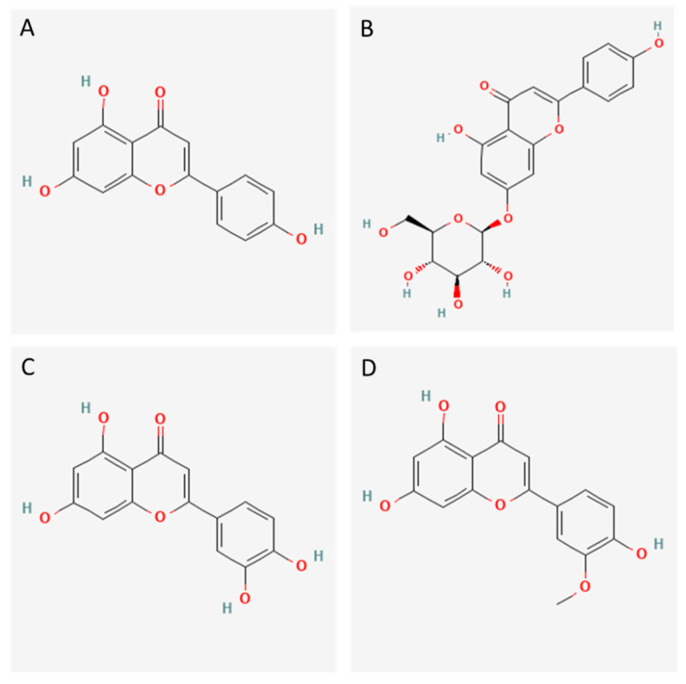
Structure of polyphenols in the bioactive fractions from the seaside arrowgrass extract. (**A**) apigenin; (**B**) apigenin-7-glucoside; (**C**) luteolin; (**D**) chrysoeriol. The structure in (**B**) corresponds to one of the isomers potentially present in the fractions. Structures were from PubChem database.

**Table 1 foods-12-03886-t001:** Total phenols content, antioxidant activity and enzymatic activity inhibitions (0.5 mg/mL, dried weight) of the sea fennel and seaside arrowgrass extracts. The determinations were performed using dried samples. Different letters in the same row indicate significant differences (*p* < 0.05).

	Sea Fennel	Seaside Arrowgrass
Total phenols content (mEq GA/g)	23.6 ± 0.1 ^a^	21.2 ± 0.1 ^b^
**Antioxidant activity**		
ABTS (mg Eq ascorbic acid/g)	20.6 ±0.4 ^a^	17.4 ± 0.6 ^b^
FRAP (mEq Mohr’s salt/g)	269.0 ± 12.0 ^a^	169.6 ± 5.8 ^b^
**Enzymatic activity inhibition**		
ACE (%)	79.6 ± 0.9 ^a^	90.2 ± 1.3 ^b^
PEP (%)	74.7 ± 3.0 ^a^	92.8 ± 2.2 ^b^
DPP-IV (%)	72.7 ± 1.1 ^a^	68.4 ± 1.2 ^b^

**Table 2 foods-12-03886-t002:** Total phenols content, antioxidant activity and enzymatic activity inhibitions of the sea fennel and seaside arrowgrass fractions. Different letters in the same row indicate significant differences among samples.

Assay	Sea Fennel				Seaside Arrowgrass			
	**SF-1**	**SF-2**	**SF-3**	**SF-4**	**SA-1**	**SA-2**	**SA-3**	**SA-4**
Total phenols content (mEq GA/g)	9.1 ± 0.97 ^a^	41.9 ± 4.2 ^c^	225.1 ± 15.7 ^g^	155.4 ± 8.4 ^f^	7.5 ± 4.0 ^a^	21.3 ± 0.7 ^b^	104.9 ± 8.4 ^d^	125.5 ± 7.7 ^e^
**Antioxidant activity**								
ABTS (mg Eq ascorbic acid/g)	8.8 ± 0.6 ^b^	32.8 ± 2.6 ^d^	177.4 ± 11.6 ^h^	124.4 ± 11.6 ^g^	4.7 ± 0.6 ^a^	12.0 ± 0.9 ^c^	53.6 ± 3.2 ^e^	90.3 ± 5.5 ^f^
FRAP (mEq Mohr’s salt /g)	248 ± 4 ^a^	636 ± 14 ^c^	3233 ± 129 ^g^	2712 ± 122 ^f^	248 ± 12 ^a^	354 ± 13 ^b^	1048 ± 14 ^d^	1701 ± 38 ^e^
**Enzymatic activity inhibition**								
ACE (%)	28.3 ± 2.1 ^a^	56.7 ± 0.3 ^c^	92.2 ± 0.1 ^g^	90.0 ± 1.1 ^f^	53.8 ± 0.7 ^b^	68.7 ± 0.7 ^d^	82.0 ± 1.0 ^e^	91.0 ± 0.2 ^f^
DPP-IV (%)	8.99 ± 2.7 ^ab^	26.8 ± 8.6 ^c^	82.1 ± 2.8 ^g^	76.1 ± 2.8 ^f^	5.56 ± 0.3 ^a^	12.3 ± 1.3 ^b^	58.2 ± 1.4 ^e^	49.7 ± 0.8 ^d^
PEP (%)	No activity	7.9 ± 2.8 ^a^	52.2 ± 2.3 ^d^	45.4 ± 0.9 ^c^	12.5 ± 5.2 ^b^	7.5 ± 1.1 ^a^	97.3 ± 0.3 ^e^	98.6 ± 0.2 ^f^

**Table 3 foods-12-03886-t003:** Polyphenolic compounds in fractions SF-3 and SF-4 from sea fennel. The relative area refers to the total compound area. Masses were obtained through negative mode analysis.

	Compound	Fraction	Experimental Mass	Calculated Mass	Error (ppm)	MS/MS Ions	Rt (min)	Relative Abundance
**Phenolic acids**								
	Hydroxybenzoic acid	SF−3	137.0238	137.0238	−3.53	92, 93, 108, 118, 136, 137	8.13	3.60
		SF−4		137.0246	−1.63		8.15	0.30
**Hydroxycinnamic acids and derivatives**								
	Caffeic acid	SF−3	179.0350	179.0352	−0.57	79, 91, 133, 134, 135	11.07	1.40
		SF−4		179.0355	0.48		11.09	2.23
	Caffeoylquinic acid	SF−3	353.0878	353.0876	−0.04	135, 161, 173, 179, 191	8.78	1.69
		SF−4		353.0883	−1.46		8.48	1.24
	Caffeoylquinic acid	SF−3	353.0878	353.0884	−1.45	135, 161, 179, 191, 217	9.55	20.26
		SF−4		353.0886	−1.79		9.57	1.36
	Caffeoylquinic acid	SF−3	353.0878	353.0875	1.30	135, 161, 173, 179, 191, 353	10.36	7.09
		SF−4		353.0886	−2.64		10.53	1.28
	Caffeoylquinic acid	SF−3	353.0878	353.0875	−2.40	161, 179, 191, 353	12.22	12.68
		SF−4		353.0875	1.11		12.25	5.48
	Dicaffeoylquinic acid	SF−3	515.1195	515.1210	−2.12	135, 161, 173, 179, 191, 335, 353	20.57	0.62
		SF−4		515.1209	−2.74		20.55	3.60
	Dicaffeoylquinic acid	SF−3	515.1195	515.1201	−0.94	135, 179, 191, 335, 353	21.37	1.43
		SF−4		515.1218	−4.32		21.37	5.58
	Dicaffeoylquinic acid	SF−3	515.1195	515.1198	−0.71	135, 173, 179, 191, 335, 353	21.93	0.14
		SF−4		515.1202	−2.22		21.92	1.65
	Dicaffeoylquinic acid	SF−3	515.1195	515.1207	−1.93	135,173,179,191,535	22.89	0.12
		SF−4		515.1212	−3.33		22.87	2.29
	Coumaroylquinic acid	SF−3	337.0929	337.0912	−4.98	93, 119, 137, 163, 173, 191,	11.73	1.87
		SF−4		337.0928	−0.06		11.73	0.57
	Coumaroylquinic acid	SF−3	337.0929	337.0948	−4.73	93, 119, 145, 163, 173, 191	12.59	14.40
		SF−4		337.0935	−1.70		12.59	13.91
	Coumaroylquinic acid	SF−3	337.0929	337.0929	−4.46	93, 119, 137, 163, 173, 191	13.00	1.83
		SF−4		337.0939	−2.29		13.00	1.46
	Coumaroylquinic acid	SF−3	337.0929	337.0927	−0.87	137, 145, 163, 173, 191	15.19	12.08
		SF−4		337.0939	−5.92		15.19	13.96
	Feruloylquinic acid	SF−3	367.0350	367.1042	−1.84	134, 173, 191, 193	14.19	6.28
		SF−4		367.1044	−2.18		14.22	8.77
	Feruloylquinic acid	SF−3	367.1035	367.1052	−4.41	134, 173, 191, 193	16.34	6.28
		SF−4		367.1028	2.14		16.33	4.49
	Coumaroyl hexoside	SF−3	325.0929	325.0938	−2.59	119, 120, 163, 164, 165, 325	9.77	2.86
**Coumarins and derivatives**								
	Esculetin	SF−3	177.0188	177.0192	−0.36	105, 107, 121, 133, 149	10.63	1.10
		SF−4		177.0918	−2.57		10.63	0.38
**Flavonols and derivatives**								
	Quercetin	SF−4	301.0354	301.0356	−0.42	65, 83, 107, 121, 151, 179, 301	28.91	1.16
	Quercetin hexoside	SF−3	463.0876	463.0883	−0.07	151, 179, 243, 255, 271, 300, 301	18.41	0.83
		SF−4		463.089	−1.82		18.42	6.47
	Quercetin hexoside	SF−3	463.0876	463.0895	−2.72	149, 179, 271, 300, 301	18.56	0.41
		SF−4		463.0892	−1.93		18.65	5.68
	Rutin	SF−3	609.1461	609.1455	1.43	151, 271, 300, 301	18.02	2.37
		SF−4		606.1455	1.43		18.02	7.08
	Myricetin	SF−4	317.0303	317.0298	1.56	109, 125, 151, 152, 163, 179, 227, 271, 317	14.03	2.10
**Flavones and derivatives**								
	Apigenin	SF−3	269.0455	269.0455	−0.42	107, 117, 149, 151, 225, 269	33.02	0.24
		SF−4		269.0466	−3.77		33.13	2.61
	Apigenin hexoside	SF−3	431.0984	431.0995	−1.82	270, 283, 311, 312, 341	17.90	0.17
		SF−4		431.0982	0.51		17.89	0.31
	Apigenin hexoside	SF−3	431.0984	431.0984	0.58	No fragment detected	18.08	0.05
		SF−4		431.0993	−2.42		18.01	0.09
	Diosmin	SF−3	607.1668	607.1681	−1.98	284, 285, 299, 300, 301	22.10	0.10
		SF−4		607.1667	0.66		22.10	1.75
	Luteolin	SF−3	285.0405	285.0408	0.04	107, 133, 149, 151, 175, 199, 217	28.76	0.07
		SF−4		285.0401	1.33		28.74	3.54
	Chrysoeriol	SF−3	299.0561	299.0556	1.12	256, 284, 299	33.66	0.03
		SF−4		299.0570	−2.83		33.70	0.48
	Diosmetin	SF−4	299.0561	299.0552	2.53	107, 133, 151, 227, 239, 255, 256, 284	33.81	0.18

**Table 4 foods-12-03886-t004:** Summarized polyphenol composition of fractions SF-3 and SF-4 from sea fennel, expressed in percentage.

Compounds	SF-3 (%)	SF-4 (%)
**Phenolic acids** (Hydroxybenzoic acid)	**3.60**	**0.30**
**Hydroxycinnamic acids and derivatives**	**91.03**	**67.87**
Caffeic acid	1.40	2.23
Caffeoylquinic acids	41.72	9.36
Dicaffeoylquinic acids	2.31	13.12
Coumaroylquinic acids	30.18	29.90
Feruloylquinic acids	12.56	13.26
Coumaroyl hexosides	2.86	0.00
**Coumarins and derivatives** (Esculetin)	**1.10**	**0.38**
**Flavonols and derivatives**	**3.61**	**22.49**
Quercetin	0.00	1.16
Quercetin hexosides	1.24	12.15
Rutin	2.37	7.08
Myricetin	0.00	2.10
**Flavones and derivatives**	**0.66**	**8.96**
Apigenin	0.24	2.61
Apigenin hexosides	0.22	0.40
Diosmin	0.10	1.75
Luteolin	0.07	3.54
Chrysoeriol	0.03	0.48
Diosmetin	0.00	0.18

**Table 5 foods-12-03886-t005:** Polyphenolic compounds in fractions SA-3 and SA-4 from seaside arrowgrass. The relative area refers to the total area of the compounds. Masses were obtained in the negative mode analysis.

	Compound	Fraction	Experimental Mass	Calculated Mass	Error (ppm)	MS/MS Ions	Rt (min)	Relative Abundance
**Phenolic acids**								
	Hydroxybenzoic acid	SA−3	137.0238	137.0247	−1.78	92, 93, 108, 118, 136, 137	8.15	0.36
		SA−4		137.0251	−1.02		8.17	0.01
**Hydroxycinnamic acids and derivatives**								
	Caffeic acid	SA−3	179.0350	179.0359	−3.56	91, 133, 134, 135	11.05	0.91
**Coumarins and derivatives**								
	Esculetin	SA−3	177.0188	177.0190	0.51	105, 107, 121, 133, 149	10.58	0.87
**Flavones and derivatives**								
	Apigenin	SA−3	269.0455	269.0458	−0.71	107, 117, 143, 151, 159, 227, 241	33.08	6.96
		SA−4		269.0452	0.98		33.05	52.41
	Apigenin−hexoside	SA−3	431.0984	431.0977	1.58	270, 283, 311, 312, 341	17.89	29.85
		SA−4		431.0982	−1.19		17.90	0.04
	Apigenin−hexoside	SA−3	431.0984	431.0988	−0.67	270, 283, 311, 312, 341	18.02	50.62
		SA−4		431.0994	−2.23		18.02	0.07
	Luteolin	SA−3	285.0405	285.0401	1.45	107, 133, 149, 151, 175, 199, 217	28.68	8.48
		SA−4		285.0399	1.92		28.66	34.66
	Chrysoeriol	SA−3	299.0561	299.0556	1.12	256, 284, 299	33.66	1.95
		SA−4		299.0563	−0.78	256, 284, 285, 299	33.67	12.81

**Table 6 foods-12-03886-t006:** Summarized polyphenol composition of SA-3 and SA-4 from seaside arrowgrass, expressed in percentage.

Compounds	SA-3 (%)	SA-4 (%)
**Phenolic acids** (Hydroxybenzoic acid)	**0.36**	**0.01**
**Hydroxycinnamic Acids and derivatives** (Caffeic acid)	**0.91**	**0.00**
**Coumarins and derivatives** (Esculetin)	**0.87**	**0.00**
**Flavones and derivatives**	**97.86**	**99.99**
Apigenin	6.96	52.41
Apigenin hexoside	80.47	0.11
Luteolin	8.48	34.66
Chrysoeriol	1.95	12.81

## Data Availability

The datasets generated for this study are available on request from the corresponding author.

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
