# Peer review of "Identification of Polyphenols in Sea Fennel (Crithmum maritimum) and Seaside Arrowgrass (Triglochin maritima) Extracts with Antioxidant, ACE-I, DPP-IV and PEP-Inhibitory Capacity"

_foods, 2023, doi:10.3390/foods12213886_

Round 1

Reviewer 1 Report

The authors wrote high quality manuscript, with very interesting results and significant content. However, some comments should be addressed: 

1. Please, include some key findings in the abstract section.

2. The Introduction part from line 78-110 is too long. Please, exclude well-known scientific facts.

3. Moisture content of the dried material is missing.

4. The letters that indicate significant differences in Table 1 should be moved after the SD values.

Author Response

Thanks for your useful comments. We have improved the manuscript according to your suggestions.

  1. Please, include some key findings in the abstract section.

Dear referee, we have modified some paragraphs to be more concise, and included results in the abstract.

  1. The Introduction part from line 78-110 is too long. Please, exclude well-known scientific facts.

Dear referee, the paragraph has been shortened

  1. Moisture content of the dried material is missing.

Dear referee, we cannot calculate the moisture content of a dried sample (extract). We expect that the moisture content was very low, as the extracts were a powder. We know the original moisture content of the plants, it was 87.1% (sea fennel) and 90.1 (seaside arrograss). We did not consider to include the composition of the plants in this article as it can be found in our previous article, Sanchez-Faure et al. (reference 13 in the manuscript).

  1. The letters that indicate significant differences in Table 1 should be moved after the SD values.

Dear referee, it has been done.

Reviewer 2 Report

Manuscript :Identification of polyphenols in sea fennel (Chritmum mariti- 2 mum) and seaside arrowgrass (Triglochim maritima) extracts with antioxidant, ACE-I, DPP-IV and PEP-inhibitory capacity” by Marta María Calvo, María Elvira López-Caballero and Oscar Martínez is interesting. Generally part concerning Discuss is insightful, exhaustive and, in my opinion, is strong side of manuscript. However this paper need to be improved.

Materials and methods

  1. Please explain in more detail in paragraph 2.3 - how many g/mg of material was used to preparation of the extracts? Please, add the reference to the described method

  1. Paragraph 2.4. Determination of Total Phenol Content: please provide the range of the standard curve and relevant reference

  2. Paragraph 2.5. Scavenging activity of ABTS radical cation :please provide the range of the standard curve and relevant reference

  3. Paragraph 2.6. Ferric Reducing Antioxidant Power (FRAP) :please provide the range of the standard curve and relevant reference

  4. Paragraph 2.10. Chromatographic Fractionation of the Extracts: add the reference to the described method

Result and dissussion

  1. line 276: FRAP assay results were expressed in Fe2+/g, but in Table 1 and in Materials and method as (mM Mohr’s salt /g), why? Data expressed in Fe2+/g should be added in Table 1 and suitable infromation in Materials and methods

  2. line 296: explain why Authors consider that they can compare ACE inhibitory activity of

studied extracts with Loizzo et al. if they not estimated IC50 values?

  1. Why did the Authors anlyze SF-3 and SF-4 and f SA-3 and SA- 4 extracts in different way in terms of their enzyme inhibitory capacity? ? (SF-3 and SF-4 extracts were chosen to examine their ACE and DPP-IV inhibitory capacities; SA-3 and SA- 4 extracts were chosen to examine their ACE and PEP inhibitory capacities)

  1. A certain weakness of the manuscript is the lack of quantitative assessment of compounds content that were found in the extracts. Quantitative studies should be planned in future. It should be indicated in conclusion.

Author Response

Materials and methods

-    Please explain in more detail in paragraph 2.3 - how many g/mg of material was used to preparation of the extracts? Please, add the reference to the described method

            Dear referee, we used 250 grams of dried sample. This information has been included in the text, as well as the reference to the described method.

-    Paragraph 2.4. Determination of Total Phenol Content: please provide the range of the standard curve and relevant reference

            Dear referee, this information has been included in the text (0.05-0.3 mg/mL), as well as the reference to the described method.

-    Paragraph 2.5. Scavenging activity of ABTS radical cation :please provide the range of the standard curve

 Dear referee, this information has been included in the text 0.02-0.6 mg/mL), as well as the reference to the described method and relevant reference

-    Paragraph 2.6. Ferric Reducing Antioxidant Power (FRAP) :please provide the range of the standard curve and relevant reference

Dear referee, this information has been included in the text (0.1-1 mM), as well as the reference to the described method

-    Paragraph 2.10. Chromatographic Fractionation of the Extracts: add the reference to the described method

            Dear referee, we have included a reference.

Result and discussion

   - Line 276: FRAP assay results were expressed in Fe2+/g, but in Table 1 and in Materials and method as (mM Mohr’s salt /g), why? Data expressed in Fe2+/g should be added in Table 1 and suitable information in Materials and methods

    Dear referee, it is a mistake that we have corrected.

- Line 296: explain why Authors consider that they can compare ACE inhibitory activity of studied extracts with Loizzo et al. if they not estimated IC50 values?

In an enzyme inhibition curve, 90% inhibition values obtained with 0.5 mg/ml inhibitor normally lie in a constant inhibition zone, in which the inhibition of higher or lower inhibitor concentrations is constant. The IC50 value that could correspond to these samples, in theory, would be more or less equal to or lower than the values reported by Loizzo et al. Logically, this is a theoretical value and we cannot demonstrate it as we have not determined IC50 values. Thus, we have changed the paragraph and deleted the sentence mentioned.  

- Why did the Authors anlyze SF-3 and SF-4 and f SA-3 and SA- 4 extracts in different way in terms of their enzyme inhibitory capacity? ? (SF-3 and SF-4 extracts were chosen to examine their ACE and DPP-IV inhibitory capacities; SA-3 and SA- 4 extracts were chosen to examine their ACE and PEP inhibitory capacities).

Dear referee, we have included all the results obtained for all the bioactivities analysed and for all the fractions (see Table 2). We have also included an statistical analysis. It has led to change the text in section 3.2.    

-    A certain weakness of the manuscript is the lack of quantitative assessment of compounds content that were found in the extracts. Quantitative studies should be planned in future. It should be indicated in conclusion.

Thanks a lot for your comment. We have included it in the conclusion

Reviewer 3 Report

The originality of the study and the novelty it brings to the field is of actuality. The purpose of the article and its significance are stated clearly. Several statements are repeated throughout the introduction section, this must be avoided to make the paper more comprehensive.

Mention the reagents' provenience.

Avoid repetition and try to focus on the main ideas regarding the results. At the same time, the Results and Discussions section could be improved by studying other papers in the field. The conclusion section is too general, I recommend the addition of relevant ideas from the current study, and a brief comment by the Authors on the strengths and weaknesses of the study presented

Author Response

-The originality of the study and the novelty it brings to the field is of actuality. The purpose of the article and its significance are stated clearly. Several statements are repeated throughout the introduction section, this must be avoided to make the paper more comprehensive.

Dear referee, thanks for your useful comment, we have tried to improve the introduction, mainly shortening a wide paragraph. We think that now it is more concise.

- Mention the reagents' provenience.

This information is now included in section 2.1. Chemicals.

- Avoid repetition and try to focus on the main ideas regarding the results. At the same time, the Results and Discussions section could be improved by studying other papers in the field. The conclusion section is too general, I recommend the addition of relevant ideas from the current study, and a brief comment by the Authors on the strengths and weaknesses of the study presented

Dear referee, we have improved the section results and discussion. We have included some papers and avoided repetitions, and the Conclusions were improved according to your suggestions.